# A Plasmid Carrying *bla*_IMP-56_ in *Pseudomonas aeruginosa* Belonging to a Novel Resistance Plasmid Family

**DOI:** 10.3390/microorganisms10091863

**Published:** 2022-09-17

**Authors:** Jessica Gómez-Martínez, Rosa del Carmen Rocha-Gracia, Elena Bello-López, Miguel Angel Cevallos, Miguel Castañeda-Lucio, Alma López-García, Yolanda Sáenz, Guadalupe Jiménez-Flores, Gerardo Cortés-Cortés, Patricia Lozano-Zarain

**Affiliations:** 1Posgrado en Microbiología, Centro de Investigaciones en Ciencias Microbiológicas, Instituto de Ciencias, Benemérita Universidad Autónoma de Puebla, Puebla 72570, Mexico; 2Programa de Genómica Evolutiva, Centro de Ciencias Genómicas, Universidad Nacional Autónoma de México, Cuernavaca 62210, Mexico; 3Departamento de Microbiología, Facultad de Ciencias Químicas, Benemérita Universidad Autónoma de Puebla, Puebla 72570, Mexico; 4Área de Microbiología Molecular, Centro de Investigación Biomédica de La Rioja (CIBIR), 26006 Logroño, Spain; 5Laboratorio Clínico. Área de Microbiología, Hospital Regional Instituto de Seguridad y Servicios Sociales de los Trabajadores del Estado, Puebla 72570, Mexico; 6Department of Microbiology and Environmental Toxicology, University of California at Santa Cruz, Santa Cruz, CA 95064, USA

**Keywords:** *Pseudomonas aeruginosa*, antimicrobial resistance, plasmids, blaIMP-56

## Abstract

*bla*_IMP_ and *bla*_VIM_ are the most detected plasmid-encoded carbapenemase genes in *Pseudomonas aeruginosa*. Previous studies have reported plasmid sequences carrying *bla*_IMP_ variants, except *bla*_IMP-56_. In this study, we aimed to characterize a plasmid carrying *bla*_IMP-56_ in a *P. aeruginosa* strain isolated from a Mexican hospital. The whole genome of *P. aeruginosa* strain PE52 was sequenced using Illumina Miseq 2 × 150 bp, with 5 million paired-end reads. We characterized a 27 kb plasmid (pPE52IMP) that carried *bla*_IMP-56_. The phylogenetic analysis of RepA in pPE52IMP and 33 *P. aeruginosa* plasmids carrying resistance genes reported in the GenBank revealed that pPE52IMP and four plasmids (pMATVIM-7, unnamed (FDAARGOS_570), pD5170990, and pMRVIM0713) were in the same clade. These closely related plasmids belonged to the MOB_P11_ subfamily and had similar backbones. Another plasmid (p4130-KPC) had a similar backbone to pPE52IMP; however, its RepA was truncated. In these plasmids, the resistance genes *bla*_KPC-2_, *bla*_VIM_ variants, *aac(6′)-Ib4*, *bla*_OXA_ variants, and *bla*_IMP-56_ were inserted between *phd* and resolvase genes. This study describes a new family of plasmids carrying resistance genes, with a similar backbone, the same RepA, and belonging to the MOB_P11_ subfamily in *P. aeruginosa*. In addition, our characterized plasmid harboring *bla*_IMP-56_ (pPE52IMP) belongs to this family.

## 1. Introduction

*Pseudomonas aeruginosa* is an opportunistic pathogen causing nosocomial infections such as ventilator-associated pneumonia, urinary tract infections, blood-associated infections, and skin and soft tissue infections [1,2,3]. Infections with this microorganism are challenging to treat due to its natural resistance and the accelerated emergence of strains resistant to almost all antibiotics, including carbapenems (last-resort treatments) [1]. Therefore, the World Health Organization in 2017 included *P. aeruginosa* in the critical-level priority pathogens group, along with *Acinetobacter baumannii* and carbapenem-resistant *Enterobacteriaceae* [4].

The main mechanisms involved in carbapenem resistance in *P. aeruginosa* are loss of the OprD porin, overexpression of efflux pumps, overexpression of the chromosomal beta-lactamase AmpC, and acquisition of carbapenem resistance genes through mobile genetic elements such as plasmids [5,6]. Class A, B, and D carbapenemases have been reported in *P. aeruginosa* strains; however, class B (Metallo-β-lactamases) are the most prevalent, of which VIM enzyme is the most frequent, followed by IMP [7]. There are currently 90 IMP metalloenzyme variants reported in the beta-lactamase database [8], including IMP-56, which varies from IMP-18 in a single amino acid (Ser214Gly) (99.59% identity). IMP-18 was reported for the first time in a *P. aeruginosa* isolated from bronchial aspirate in the southwestern United States [9]. IMP-18 has been found in Mexico, Puerto Rico, and France [10,11,12,13,14,15] and is commonly associated with chromosomic class 1 integrons containing other antibiotic resistance genes such as acetylases and oxacillinases [15,16,17].

The *bla*_IMP-56_ sequence was reported in January 2016 in a *P. aeruginosa* strain from Guatemala (GenBank accession no. NG_049218.1) and by our research group (GenBank accession no. KY646161) in a strain from México [15]. In addition, this variant was reported in a class 1 integron and located by Southern blot in a plasmid [15]. IMP and VIM Metallo-β-lactamases are the most detected carbapenemases in plasmids of *Pseudomonas aeruginosa* [18,19,20,21,22] as well as the KPC-2 serine carbapenemase [3,23,24]. The plasmid sizes carrying these carbapenemases vary widely; the range reported in GenBank is between 2 kb and 500 kb. According to the GenBank database, one to two plasmids in *P. aeruginosa* strains have been reported, although five strains carry up to four plasmids (GenBank accession no. CP077988.1, CP077971.1, CP086010.1, LT969520.1, CP065412.1).

Until now, fourteen incompatibility groups have been recognized in *P. aeruginosa* plasmids (IncP-1–IncP-14) [25]. Incompatibility groups IncP-2, IncP-6, and IncP-7 are the most associated with plasmids carrying carbapenem resistance genes in *P. aeruginosa* [7]. However, contrary to *Enterobacteriaceae* and *Acinetobacter*, a PCR-Based Replicon Typing (PBRT) to classify *P. aeruginosa* plasmids has not been reported [26,27]. On the other hand, the degenerate primer MOB typing (DPMT) method, which allows the classification of gamma-proteobacteria plasmids [28], has been used in some studies to classify *P. aeruginosa* plasmids [15].

Previous studies have reported different plasmid-borne *bla*_IMP_ variants in clinical strains of *P. aeruginosa*. For example, *bla*_IMP-4_ was reported in a 51,207 bp conjugative plasmid and possesses the conserved IncN1-type backbone regions (*repA*, iterons, conjugal transfer genes, and maintenance genes) [20]. On the other hand, *bla*_IMP-9_ and *bla*_IMP-45_ were reported in 500,839 and 513,322 bp megaplasmids, respectively, with a highly similar backbone [18,29].

Our laboratory reported the *P. aeruginosa* strain PE52 isolated from a Mexican hospital and harboring a class 1 integron carrying *bla*_IMP-56_, detected in a plasmid by Southern blot [15]. Therefore, in this study, we aimed to characterize the plasmid carrying *bla*_IMP-56_ in *P. aeruginosa* strain PE52.

## 2. Materials and Methods

### 2.1. Characteristics of P. aeruginosa PE52

For the characterization of the *bla*_IMP-56_-carrying plasmid, we used the *P. aeruginosa* strain PE52, which was isolated from the urinary culture of a patient admitted to the Hospital Regional I.S.S.S.T.E de Puebla, Mexico. The mechanisms of resistance and the presence of plasmids were previously described [15]. Furthermore, this strain carried *bla*_IMP-56_ in a new class 1 integron arrangement and we experimentally observed that *bla*_IMP-56_ was probably present in a plasmid [15].

### 2.2. Genome Sequencing

Genomic DNA from strain PE52 was extracted using the Wizard^®^ Genomic DNA Purification Kit (Promega Corporation, Madison, WI, USA). The whole-genome sequencing was executed by Ilumina Miseq 2 × 150 bp, with 5 million paired-end reads in the SNPSaurus Genomics to Genotype [30].

### 2.3. Plasmid Analysis

The quality of the paired-end reads was measured by FastQC version 3.9.0 [31]. BLASTn [32], PLACNETw [33], and the plasmid SPAdes version 3.9.0 (St. Petersburg State University, St Petersburg, Russia) [34] were used to differentiate the nodes of plasmids. The genome sequences were annotated with Rapid Annotations using Subsystem Technology (RAST) [35]. Plasmids were drawn using Proksee [36]. Finally, oriTfinder was used to identify the origin of the transfer site in pPE52IMP [37].

### 2.4. Plasmid Characterization

The MOBScan web application [38] was used to identify relaxases and classify the plasmids into any of the nine MOB families. For in silico classification by the replicon method, we used PlasmidFinder [39].

To determine the phylogenetic relationship of RepA protein in pPE52IMP and plasmids from *P. aeruginosa*, we analyzed the RepA of plasmids carrying resistance genes and constructed a phylogenetic tree. A total of 164 nucleotide sequences of complete and partial plasmids from the GenBank database were obtained (until October 2021). The plasmid sequences were annotated with Rapid Annotations using Subsystem Technology (RAST) [35] and ResFinder version 4.1 [40] for the detection of antibiotic resistance genes in all plasmids. It is essential to point out that plasmids that did not carry resistance genes were not included. RepA amino acid sequences of the plasmids carrying resistance genes were searched in the annotations using the keywords “replicase”, “repA”, and “helix-turn-helix domain-containing protein”, and their replicase A domains were corroborated with Pfam [41]. In addition, RepA with premature stop codons or ORF changes were discarded. Finally, 33 RepA proteins of plasmids (Appendix A) (including pPE52IMP) were used to construct the phylogenetic tree. The Molecular Evolutionary Genetics Analysis tool, MEGA version 11.0.10 [42], was used to infer RepA proteins’ phylogeny using the UPGMA method (the parameters used were amino acid substitution type, no. of differences method, and 100 bootstrap replicates).

### 2.5. Comparative Analysis of Plasmids Obtained from GenBank and pPE52IMP

For the comparative analysis of plasmids, we selected the complete sequences of the plasmids that shared 100% identity with *repA* of pPE52IMP and were in the same clade in the phylogenetic tree. To align and compare the sequences, we used MAUVE version 20150226 [43] and CLC Sequence Viewer version 8.0 (CLC bio A/S, Aarhus N, Denmark). To represent the comparison of plasmids, EASYFIG 2.2.5 was used [44].

## 3. Results

### 3.1. Structural Features of the pPE52IMP Plasmid

Whole-genome sequencing revealed the presence of a single plasmid, pPE52IMP, that carries the *bla*_IMP-56_ variant (GenBank accession no. CP102481.1). The pPE52IMP plasmid had a size of 27,635 bp, 39 open reading frames (ORFs), and guanine–cytosine (G+C) content of 62.2%. Moreover, 32 of the 39 open reading frames had a predicted function: 1 of replication, 6 of stability, 7 of transfer, 13 of adaptation, and 5 transposon-related genes. We could not determine the functional domain of seven hypothetical proteins (Figure 1).

The transfer module consisted of the genes *traJ, traK, trbL, trbK, trbJ, virB4*, and a relaxase *traI* belonging to the MOB_P11_ subfamily. The *oriT* was located upstream of *traK* and consisted of 113 bp. The stability module involves the partitioning genes *parA* and *parC*; however, the *parB* gene was not found. In addition, the toxin–antitoxin genes *phd/doc* and *krfA* gene were identified. The *repA* was part of the replication module, and no iterons or replication origins close to the *repA* gene were found (Figure 1).

The adaptation module contained a class 1 integron carrying *bla*_IMP-56_, *aadA1*, and *bla*_OXA-2_ genes. In addition, the Tn3 family transposon carrying a mercury resistance operon (*merR, merT, merP, merA, merD*, and *merE* genes) was located (Figure 1).

### 3.2. Phylogenetic Analysis of RepA

To infer a possible phylogenetic relationship between pPE52IMP and other plasmids from *P. aeruginosa*, we used RepA to build a phylogenetic tree. For analysis, we included the amino acid sequence of 33 RepA from plasmids carrying antibiotic resistance genes (Appendix A) (including RepA of pPE52IMP). The analysis showed a wide diversity of replicases among *P. aeruginosa* plasmids grouped in 11 clades (Figure 2). Furthermore, RepA proteins of plasmids with the same incompatibility group were clustered in the same clade such as IncP-2 (pOZ176, pJB37, pPUV-1), IncP-6 (C79, p10265-KPC, pCOL-1), and IncP-7 (p1160-VIM and pNK546b); however, the incompatibility group of one plasmid within the IncP-7 clade (unnnamed1 P8W) was not reported (Figure 2). On the other hand, it is important to note that RepA proteins from pPE52IMP, pMATVIM-7, unnamed1 (FDAARGOS_570), pD5170990, and pMRVIM0713 were in the same clade (Figure 2).

### 3.3. Comparative Analysis of pPE52IMP and Plasmids with Same RepA and Similar Structure

pPE52IMP structure was compared with closed plasmids from the GenBank, which clustered in the same clade of the RepA phylogenetic tree (Figure 2): pMATVIM-7 (GenBank accession no. AM778842.1), plasmid unnamed (GenBank accession no. CP033834.1), pD5170990 (GenBank accession no. KX169264.1), and pMRVIM0713 (GenBank accession no. KP975076.1). In addition, we found the plasmid p4130-KPC (GenBank accession no. MN336501.1) in GenBank, which had a similar backbone to pPE52IMP. However, it was not included in the phylogenetic analysis because its RepA was truncated, but it was incorporated in the comparative analysis of the structure (Figure 2). These plasmids ranged from 24 kb to approximately 58 kb, and were isolated in the USA, Brazil, and France. The characteristics of these plasmids are shown in Appendix A.

The comparative analysis showed that these six plasmids shared a similar backbone, including genes for replication (*repA*), partition (*parA*, *parC*), and transfer (*tra* and *virB4*); however, we found some differences. *traJ*, *traK*, and *kfrA* genes were absent of pD517099. *trbJ* gene in p4130-KPC was interrupted by a transposon, the N-terminus of TraK in pMAT-VIM-7 is absent, and RepA of p4130-KPC lacks the C-terminus (Figure 3).

Figure 3 illustrates that the variable region was found downstream of the *phd* gene and upstream of the resolvase gene and consisted of genes for adaptation such as carbapenemases type *bla*_IMP-56_ (pPE52IMP) and *bla*_VIM-6_ (plasmid unnamed and pMRVIM0713) carried by a class 1 integron, *bla*_VIM-7_ (pMATVIM-7) carried by a partial class 1 integron, *bla*_KPC-2_ (pD5170990) carried by a transposon, and *bla*_OXA-779_, *bla*_OXA-732_, and *bla*_KPC-2_ brought by a class 1 integron and a transposon, respectively (p4130-KPC) (Figure 3).

As previously mentioned, pPE52IMP was classified into the MOB_P11_ subfamily [28] but was not classifiable by replicon typing. In silico analysis revealed that plasmids pMATVIM-7, unnamed (FDAARGOS_570), pD5170990, pMRVIM0713, and p4130-KPC were classified into the MOB_P11_ subfamily but were not classifiable according to the replicon typing scheme [26,45]. These plasmids shared some characteristics, such as having a same replicase and similar backbone, and were classified as MOB_P11_, but were not classifiable by the incompatibility group.

### 3.4. Plasmids with Similar Backbone as pPE52IMP Present in Other Bacterial Genera

By searching GenBank using the *repA* gene from pPE52IMP, we found two plasmids with the same *repA* and similar backbones in *Achromobacter ruhlandii* (plasmid p138R) and *Serratia marcescens* (plasmid pSMC1). The sizes of the plasmids were 34 and 41.5 kb and they were isolated from Argentina and Japan, respectively (Appendix A). Comparing the complete sequence of the plasmids, we determined that these two plasmids shared a conserved backbone with pPE52IMP. Furthermore, we found that almost all backbone genes shared 100% identity and coverage, except for *repA* of p138R, which was truncated, and *traI* of pSMC1, which shared 97.65% nucleotide similarity and 100% coverage. In addition, the variable region of these plasmids carried different carbapenemases (*bla*_IMP-1_, *bla*_CMY-8_) and other resistance genes such as *aac(6′)-Ib4* and *aadA2*, commonly found in enterobacteria (Appendix A).

## 4. Discussion

The emergence of beta-lactamases with activity against carbapenems has compromised the clinical utility of this class of antibiotics [46]. In *P. aeruginosa*, class A and B β-lactamases with carbapenemase activity are reported, including VIM, IMP, SPM, NDM, GIM, GES, and KPC [47,48]. IMP, VIM, NDM, and GES types comprise several variants, whereas only one variant for SPM-1 and GIM-1 have been reported [49]. These enzymes are carried in plasmids, integrons, and transposons, which play an important role in their dissemination [49]. Recently, carbapenemases mobilized by mobile genetic elements in *Pseudomonas aeruginosa* were reviewed and it was found that *bla*_KPC-2_, *bla*_VIM-1_, and *bla*_IMP-45_ are carried by plasmids belonging to different incompatibility groups [7]. In addition, other carbapenemases such as *bla*_VIM-2_, *bla*_IMP-6_, and *bla*_IMP-9_ are carried by plasmids [18,21,50,51]. Little is known about *P. aeruginosa* plasmids and their role in resistance gene dissemination; therefore, characterizing plasmids will help better understand this dissemination mechanism.

In this work, we determined the structure of the plasmid pPE52IMP carrying *bla*_IMP-56_ (Figure 1), finding that it has lower G+C content (62.2%) than the *P. aeruginosa* chromosome (approximately 66.6%) [52]; however, it is consistent with the GC content reported in other *P. aeruginosa* plasmids (from 45.8% to 63.8%) [25]. A previous study revealed that the average GC content of plasmids was 10% lower than their host’s chromosome, which suggests that plasmids with very different GC content could not be maintained in their host [53].

The stability module comprises a partitioning system that contributes to the segregation of the plasmid, an addiction system that ensures the killing of plasmid-free cells, and multimer resolution systems that prevent the formation of plasmid multimers [54]. The partitioning system consists of ATPase (*parA*), centromere-like DNA sequence (*parC*), and DNA-binding protein (*parB*) [55]; the latter is composed of a central HTH DNA binding domain flanked by a C-terminal dimer domain and an N-terminal region necessary for protein oligomerization [56]. In the case of pPE52IMP, we found only the *parA* and *parC* genes, while the *parB* gene was absent, and none of the hypothetical proteins present in the plasmid had domains *parB*-like (Figure 1).

On the other hand, the *kfrA* gene has been shown to act as a transcriptional autoregulator and participates in plasmid stability [57,58,59], suggesting that this gene could be involved in pPE52IMP stability; however, other studies are necessary to understand how the segregation process is carried out in this plasmid. In addition, the addiction system is composed of the Doc toxin (death on curing) and Phd antitoxin (prevents host death) (Figure 1) that belongs to type II systems, where the toxin is directly blocked by the antitoxin [60]; besides, this toxin/antitoxin system plays an important role in plasmid stability persistence, programmed cell death, and stress response [61].

Conjugative plasmids carry two sets of genes; the first allows DNA processing (DNA transfer and replication (Dtr) genes), and the second is a membrane-associated mating pair formation (Mpf) complex (a form of type 4 secretion system). In contrast, mobilizable plasmids use the Mpf of another genetic element in the same cell [62,63]. The transfer module of pPE52IMP consists of IncP-like plasmid genes *traK, traJ*, and *traI*, which are essential for relaxosome formation, and the conjugative transfer genes *trbJ, trbK*, and *trbL* are involved in the formation of the Mpf system; however, the genes *traH* (chaperone activity), *traG* (coupling protein), *traA, traB, traD,* and *traE* (not essential for conjugation) and the genes *trbBCDEFGHI* (necessary for the formation of the Mpf system) are absent in pPE52IMP [64], suggesting that it could be a mobilizable plasmid. Furthermore, the lack of transconjugants in the conjugation experiment reinforces this analysis (data not shown).

Mercury operons comprise mercury resistance-conferring genes (*merEDAPTR*) and are commonly located on transposons and integrons carried by plasmids [65]. pPE52IMP carry the *mer* operon located next to the *tn21* and *tnpR* genes (Figure 1) that are part of transposable elements of the Tn3 family [66].

On the other hand, some authors have used features of the plasmid backbone to design classification schemes such as PCR-based replicon typing (PBRT) [26] and degenerate primer MOB typing (DPMT) [28] based on plasmid replication and mobility functions, respectively [67]. Plasmids of *P. aeruginosa* with a similar backbone to pPE52IMP have a MOB_P11_ subfamily relaxase according to MOB typing [28]; this is consistent with findings reported by Lopez-García [15]. The MOB_P11_ subfamily belongs to the MOBP superfamily, one of the most abundant in plasmids among gammaproteobacterial (including *Pseudomonas*) [68].

pPE52IMP and plasmids with similar backbone could not be classified by PBRT [26], which may be related to the fact that this scheme is focused on classifying plasmids from Enterobacteriaceae but not from other bacterial families. pPE52IMP does not belong to any of the 14 incompatibility groups (IncP-1 to IncP-14) described in *P. aeruginosa*; this is consistent with Shintani et al., 2015 [25], who found that only 21 of 183 *Pseudomonadales* plasmids analyzed could be classified into the IncP group. The above reflects the need to develop a technique to classify *P. aeruginosa* plasmids; however, classifying plasmids using MOB typing could help in some cases.

A classification based on replicase sequence homology was designed by Bertini for *Acinetobacter baumannii* plasmids, identifying 19 homology groups (GRs) [27]. *Rep* genes that shared at least 74% of identity were in the same group. Other authors have added more groups using the same identity criteria, reporting, to date, 33 GRs [69]. Therefore, we used similar parameters to know the distribution and behavior of RepA in pPE52IMP and plasmids of *Pseudomonas aeruginosa* reported in the GenBank (Appendix A and Figure 2). It is important to highlight that we included only plasmids carrying resistance genes in the analysis. RepA of plasmids belonging to the same incompatibility group (IncP-2, IncP-7, IncP-6) were clustered in three clades, likely because plasmids belonging to the same incompatibility group have the same or related replication/partitioning system [70]. On the other hand, the RepA of pPE52IMP and plasmids with a similar backbone were clustered together in a separated clade, indicating that they are closely related genetically and are probably a new family of plasmids.

According to the information available in the GenBank, the strains of *P. aeruginosa* and the other genera that carried plasmids similar to pPE52IMP were isolated from the USA (mainly), Brazil, France, Argentina, and Japan (Appendix A), which would indicate that these plasmids are circulating in different countries and acting as vehicles for the dissemination of antibiotic resistance genes.

Closely related plasmids commonly have a core called the “backbone” associated with plasmid-specific functions such as replication initiation, conjugation, and stability. In addition, the backbone can include virulence genes and antibiotic- and heavy metal-resistance genes that confer adaptive advantages to the bacterium [71]. In the analysis of the phylogenetic tree, we found four plasmids of *P. aeruginosa* strains, and one plasmid of a strain reported in the GenBank with a backbone similar to pPE52IMP. In addition, the plasmids had a variable region with carbapenem resistance genes such as *bla*_VIM-6_, *bla*_VIM-7_, *bla*_KPC-2_, and other beta-lactamase encoding genes such as *bla*_OXA-779_, *bla*_OXA-732_, and *bla*_OXA-10_ (Appendix A and Figure 3) carried by class 1 integrons and transposons. Our working group reported that pPE52IMP carries *bla*_IMP-56_ in a class 1 integron (GenBank accession no. KY646161) [15]; nevertheless, in this study, we report the structure of the plasmid carrying *bla*_IMP-56_, which belongs to a new family of plasmids.

Plasmids with a conserved backbone carrying resistance genes inserted into hotspot sites have been reported, and the *repA* gene serves as a hotspot in some of them [72,73,74]. However, in the plasmids analyzed, the resistance genes are inserted between *phd* and a resolvase gene so that it could be a potential hotspot for integrating the resistance genes in these plasmids, but more studies are necessary.

We also found two plasmids with backbone similar to pPE52IMP in bacteria not closely related to *P. aeruginosa*, such as p138R from *A. ruhlandii*, and pSMC1 from *S. marcescens* (Appendix A). These plasmids carried the *aadA1, aac(6′)-lb4* acetylase, *bla*_CMY-8_, and *bla*_IMP-1_ genes. These observations could indicate that plasmids of this type could be of a broad host range [73], allowing the dissemination of resistance genes between bacteria different from *P. aeruginosa*. However, transformation experiments with hosts of other bacterial genera are needed to confirm the host range of this plasmid.

## 5. Conclusions

In this study, we described a new family of plasmids carrying resistance genes with the same RepA, a similar backbone, and belonging to the MOB_P11_ subfamily in *P. aeruginosa*. In addition, we characterized the first plasmid harboring *bla*_IMP-56_ (pPE52IMP), isolated from a Mexican hospital, belonging to this family. This study contributes to understanding how these plasmids encoding carbapenemases spread among bacteria.

## Figures and Tables

**Figure 1 microorganisms-10-01863-f001:**
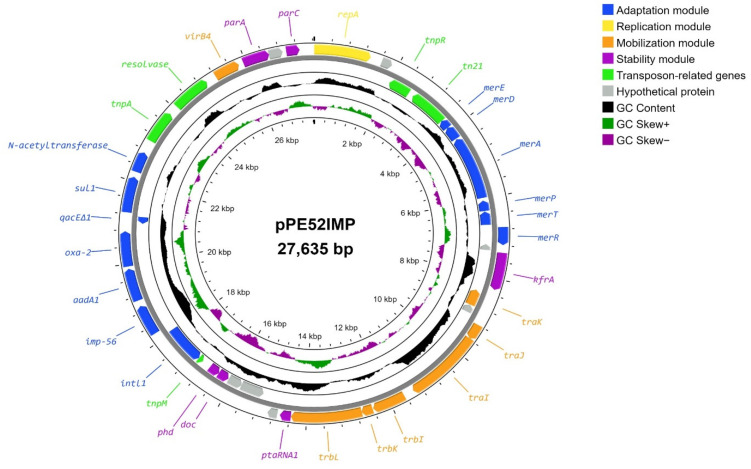
Structure of the pPE52IMP plasmid of *P. aeruginosa* strain PE52. Plasmid modules are represented with different colors. Blue: adaptation; yellow: replication; orange: mobilization; purple: stability; green: transposons; gray: hypothetical proteins. GC content, GC skew+ and GC skew– are represented in colors black, purple and green, respectively on the inner map.

**Figure 2 microorganisms-10-01863-f002:**
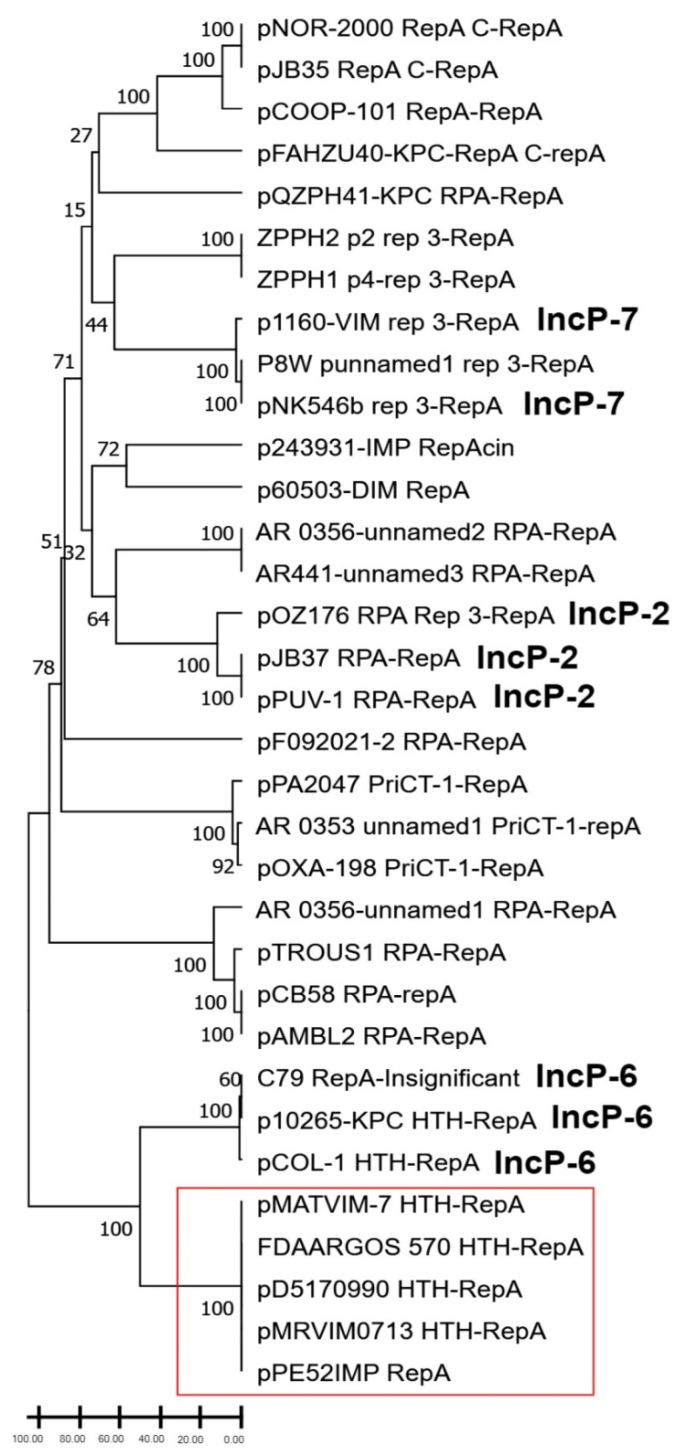
UPGMA phylogenetic tree of replicase A proteins in plasmids carrying resistance genes. The incompatibility groups of the plasmids are in bold type. Replicases in the same clade as pPE52IMP RepA are enclosed in a red box.

**Figure 3 microorganisms-10-01863-f003:**
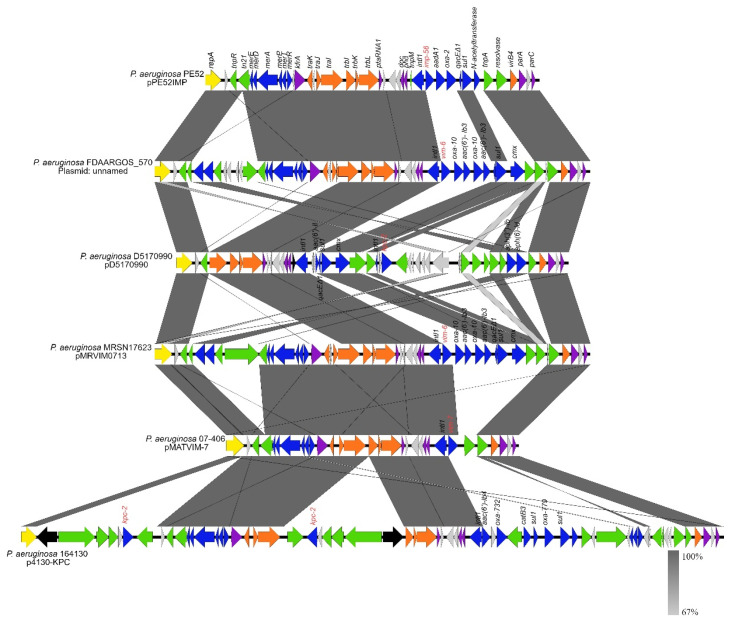
Structural comparison of plasmid pPE52IMP and plasmids with similar backbones and RepA from clinical *P. aeruginosa.* The figure was built with Easyfig. Plasmid modules were represented with different colors. Yellow: replication; purple: stability; orange: mobilization; blue: adaptation; green: transposons; gray: hypothetical proteins; black: other genes. The gray color level indicates the percentage of BLAST identity of the plasmids. Plasmid unnamed (accession no. CP033834.1); pD5170990 (accession no. KX169264.1); pMARVIM0713 (accession no. KP975076.1); pMATVIM-7 (accession no. AM778842.1); p4130-KPC (accession no. MN336501.1).

## Data Availability

*P. aeruginosa* PE52 strain was recovered from routine culture and informed patient consent was not required. The protocol to perform this study was approved by the Ethical Committee of Hospital Regional del ISSSTE, Puebla, under number 188-2018.

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
