# Peer review of "A Plasmid Carrying blaIMP-56 in Pseudomonas aeruginosa Belonging to a Novel Resistance Plasmid Family"

_microorganisms, 2022, doi:10.3390/microorganisms10091863_

Round 1

Reviewer 1 Report

The article entitled “A new plasmid the carrying carbapenem-resistance gene blaIMP-56 in Pseudomonas aeruginosa from a Mexican hospital” describes a new AMR plasmid present in an isolate of P. aeruginosa from a urine sample from a patient. The report is interesting, and the genomic study of the plasmid is well done. Unfortunately, English is not their first language, and the descriptions and details in this manuscript need a better-edited version to avoid confusing paragraphs, vague statements, and ambiguous sentences. In addition, there are several specific points. 

1-    Tittle change consideration. The title should not include where the sample was collected. Associating a country with AMR has negative consequences from the perspective of tourism, etc.  Something along the lines of “A novel plasmid family/class/group of P. aeruginosa….

2-    The authors seem to have found previously sequenced plasmids belonging to the same family of plasmids. This paper would be interesting if this fact were more central to the manuscript. 

3-    This figure should compare the plasmid family that branches together in Figure 4. Figure 4 should be figure 1, and the rest of the figures should go to supplemental material. 

4-    Table 1 also could go to supplemental material. 

Limiting the figures and tables will make this manuscript about a single plasmid more relevant. Great work!

Author Response

Puebla, Pue. México, September, 2022

Dear Reviewers

Microorganism (MDPI)

We appreciate the comments, observations, and time invested in reviewing the manuscript.

We have carefully reviewed all the reviewer suggestions. Changes and responses are given point by- point, and relevant modifications made in the manuscript paragraphs are highlighted in yellow.

The article entitled “A new plasmid the carrying carbapenem-resistance gene blaIMP-56 in Pseudomonas aeruginosa from a Mexican hospital” describes a new AMR plasmid present in an isolate of P. aeruginosa from a urine sample from a patient. The report is interesting, and the genomic study of the plasmid is well done. Unfortunately, English is not their first language, and the descriptions and details in this manuscript need a better-edited version to avoid confusing paragraphs, vague statements, and ambiguous sentences. In addition, there are several specific points. 

 Response: The manuscript was corrected and edited for the English language. In addition, a native English-speaker made a review of the punctuation, spelling, and overall style.

1. Tittle change consideration. The title should not include where the sample was collected. Associating a country with AMR has negative consequences from the perspective of tourism, etc.  Something along the lines of “A novel plasmid family/class/group of P. aeruginosa….

Response: According to the reviewer´s suggestion, we change the title to give importance to the description of the new family of plasmids. “A plasmid carrying blaIMP-56 in Pseudomonas aeruginosa belonging to a novel resistance plasmid family”, We expect that this title change is according to the recommendations of the reviewer.

2-    The authors seem to have found previously sequenced plasmids belonging to the same family of plasmids. This paper would be interesting if this fact were more central to the manuscript. 

Response: According to the appreciation made by the reviewer, we make changes in the sequence of presentation of the results and discussion, giving importance to the description of the new family of plasmids.

It is important to highlight that the conclusion was restructured to emphasize this fact.

“In this study, we described a new family of plasmids carrying resistance genes with the same RepA, similar backbone, and belonging to the MOBP11 subfamily in P. aeruginosa. In addition, we characterized the first plasmid harboring blaIMP-56 (pPE52IMP), isolated from Mexican Hospital, belonging to this family. This study could contribute to understanding how these plasmids encoding carbapenemases spread among bacteria”.

3-    This figure should compare the plasmid family that branches together in Figure 4. Figure 4 should be figure 1, and the rest of the figures should go to supplemental mat

Response: The reviewer’s suggestion seems very important to us, so we changed the article in the sequence of results and discussion. We moved figure 4 to figure 2 because we considered that the plasmid description is important (figure 1). Also, we conserve figure 3 (Comparison of the structure of plasmid pPE52IMP and plasmid with similar backbones and RepA of clinical P. aeruginosa) because, in this figure, we compare another plasmid not included in the phylogenetic tree for having a truncated RepA, and the backbone of all the plasmids can be visualized. Other figures and tables were in the supplementary material.

 4-    Table 1 also could go to supplemental material. Limiting the figures and tables will make this manuscript about a single plasmid more relevant.

 Response: The tables were sent to supplementary material.

Reviewer 2 Report

Identification and characterization of new plasmids are very critical in curtailing antimicrobial resistance in bacterial pathogens. Gomez-Marinez  and colleagues have identified a new a plasmid, blaIMP-56 from Pseudomonas. The authors also have done extensive characterization to understand the phenotypic and genotypic properties of the plasmid. 

Author Response

Puebla, Pue. México, September, 2022

 Dear Reviewer

Microorganism (MDPI)

 We appreciate the comments, observations and time invested in reviewing the manuscript.

We have carefully reviewed all the reviewer suggestions. Changes and responses are given point by- point, and relevant modifications made in the manuscript paragraphs are highlighted in yellow.

  1. English language and style are fine/minor spell check required

Response: The manuscript was corrected and edited for the English language. In addition, a native English-speaker made a review of the punctuation, spelling, and overall style

  1. Identification and characterization of new plasmids are very critical in curtailing antimicrobial resistance in bacterial pathogens. Gomez-Marinez and colleagues have identified a new a plasmid, blaIMP-56 from The authors also have done extensive characterization to understand the phenotypic and genotypic properties of the plasmid.

Response: We appreciate the time taken to review the manuscript.

According to the reviewer suggestions, and the changes made, we hope that this manuscript will be considered for publication

Best regards,

Patricia Lozano Zarain Ph.D.

(Corresponding author)

Centro de Investigaciones en Ciencias Microbiológicas

Instituto de Ciencias. Benemérita Universidad Autónoma de Puebla

Complejo de Ciencias, Edif. IC-11, Ciudad Universitaria Colonia San Manuel, CP 72570, Puebla, Mexico Phone: + 52-222-2295500 ext. 2543

Round 2

Reviewer 1 Report

Great work!